# Genetic Characterization and Mating Disruption in *Spodoptera* Species, a Case Study on *Spodoptera frugiperda* (Lepidoptera, Noctuidae): A Systematic Review

**DOI:** 10.3390/insects16111176

**Published:** 2025-11-18

**Authors:** Carla Tavares, Sílvia Catarino, António Mexia, Elsa Borges da Silva, Filipa Monteiro

**Affiliations:** 1Linking Landscape, Environment, Agriculture and Food Research Centre (LEAF), School of Agriculture (ISA), University of Lisbon, Tapada da Ajuda, 1349-017 Lisbon, Portugal; amexia@isa.utl.pt; 2Forest Research Centre (CEF), School of Agriculture (ISA), University of Lisbon, Tapada da Ajuda, 1349-017 Lisbon, Portugal; elsasilva@isa.ulisboa.pt; 3DGASP-MAA—General Direction of Agriculture, Forestry and Livestock-Ministry of Agriculture and Environment, Praia PC 278, Santiago Island, Cape Verde; 4Departamento de Alterações Climáticas, Agência para o Clima (ApC), I. P., Rua de “O Século” 51, 1200-433 Lisbon, Portugal; 5Associate Laboratory TERRA, School of Agriculture (ISA), University of Lisbon, Tapada da Ajuda, 1349-017 Lisbon, Portugal; 6CE3C, Centre for Ecology, Evolution and Environmental Changes & CHANGE—Global Change and Sustainability Institute, Faculty of Sciences, University of Lisbon, Campo Grande, 1749-017 Lisbon, Portugal

**Keywords:** PRISMA, strain identification, semiochemical, integrated pest management

## Abstract

The *Spodoptera* genus includes several species whose caterpillars cause major crop losses, notably *Spodoptera frugiperda* (J.E. Smith, 1797), which has spread rapidly worldwide and is difficult to control. One promising approach to managing these pests is the use of synthetic substances that attract them and disrupt their reproduction and population dynamics. The success of this approach depends on understanding the biology of the pest and its ecology and behaviour under various environmental conditions. This review summarizes studies on the genetic identification of *Spodoptera* species and the use of attractive synthetic substances for their control, showing that their genetic information helps us to better understand these pests, as well as the effectiveness of various methods. The findings highlight the need to standardize attractive formulations and adapt management strategies to local agroecological conditions, combined with genetic knowledge in supporting the sustainable control of *S. frugiperda*.

## 1. Introduction

The genus *Spodoptera* (Lepidoptera: Noctuidae) includes various species of major agricultural pests found worldwide [1,2]. Notable species include *Spodoptera frugiperda* (J.E. Smith, 1797), *Spodoptera exigua* (Hübner, 1808), *Spodoptera exempta* (Walker, 1857), *Spodoptera litura* (Fabricius, 1775), and *Spodoptera littoralis* (Boisduval, 1833). According to FAO [3], there is evidence that some *Spodoptera* species are spreading beyond their natural environment due to ongoing global climate change. Among these, *S. frugiperda*, an invasive pest native to America, first invaded Africa in 2016, specifically Nigeria and São Tomé and Príncipe, and rapidly spread to other African countries [4]. Subsequently, it spread to Asia in 2020, emerging as a significant global problem [5].

*Spodoptera frugiperda* is recognized as a major pest threatening agriculture worldwide, causing significant damage to important crops such as maize and rice, particularly in newly invaded regions. It poses a critical challenge to food security, especially in areas where maize is a staple crop and resources for pest management are limited, as is often the case in the African context. The annual losses caused by this pest on maize production crop are estimated at USD 300–500 million in the U.S., USD 1.1–4.7 billion in sub-Saharan Africa (up to USD 13 billion across major crops), and 4.1–17.7 million tons of maize only in 12 key African countries [6,7,8,9]. Its destructive capacity is due to its biological characteristics, which include a wide host range, high migratory capacity, strong fertility, and the absence of a diapause mechanism, i.e., population outbreaks largely depend on environmental conditions, in addition to their ability to develop resistance to pesticides [9,10,11,12]. While chemical control remains the main method used against *S. frugiperda*, its effectiveness is limited, with reduced susceptibility and field-evolved resistance reported in both native and invaded regions [13,14,15]. Resistance to *Bacillus thuringiensis* (Bt) toxins has also emerged despite the widespread use of transgenic Bt crops for *S. frugiperda* control [15]. These developments highlight the challenges of relying on chemical and Bt-based control methods for the long-term management of the pest. Thus, there is an urgent need to explore alternative environmentally friendly control methods, reducing reliance on chemical pesticides, minimizing risks to the environment, and ensuring the sustainable control of *S. frugiperda* infestation.

The sustainable management of *S. frugiperda* requires integrated control methods, including the use of semiochemicals, such as sexual pheromones, which have been effective in controlling some Lepidoptera species [16,17]. Pheromones are highly effective in integrated pest management because males are highly mobile and respond to low concentrations of species-specific pheromones. They also reliably monitor adult flight activity, facilitating the targeting of adults, which are more susceptible to treatments than larvae. Thus, pheromones can also be used to monitor the distribution of *S. frugiperda*, supporting early warning and control efforts in invaded areas. Several authors report that mating disruption, in which synthetic sexual pheromones are strategically released to interfere with reproductive biology, reduces pest populations to acceptable levels and provides a targeted, environmentally friendly alternative to chemical control, representing an important method for integrated pest management [18,19,20,21]. While the use of sexual pheromones has proven effective in the control of certain Lepidoptera species [16,17], the efficacy of this approach for *S. frugiperda* control depends on selecting the appropriate pheromone blend according to the biological characteristics of populations in each region [22]. For instance, pheromone blends that have proven successful in North America and Europe performed poorly in South America due to geographical variability [23]. Sexual pheromones, typically species-specific, are emitted by females to attract males for mating, with their composition varying based on insect strain populations [24]. Therefore, to maximize the efficacy of pheromones, it is essential to identify the strains, determine their pheromone composition, and understand the responses of male moths to pheromone blends that reflect the strain variability found in crop fields [22].

To date, within the *Spodoptera* genus, clear evidence of host-associated strains has been reported only in *S. frugiperda*, which consists of two genetically distinct yet morphologically indistinguishable strains: the corn strain (C-strain) and the rice strain (R-strain) [25]. The identification of these strains is restricted to molecular methods, including the use of mitochondrial polymorphisms and a small number of sex-linked nuclear markers, which limits strain identification based on host plant association [26,27]. Despite genetic differentiation, evidence of hybridization between the two strains has been observed, suggesting incomplete reproductive isolation and gene flow, with mating occurring despite host preferences [28]. In contrast, no host-associated strains have been reported in other *Spodoptera* species, such as in *S. exigua* and *S. litura*, where the genetic structure appears to be more closely related to geographic population differentiation rather than host association [29,30,31]. Still, *S. frugiperda*’s genetic variability and migratory behaviour in Africa remain uncertain. Previous studies suggest that Western Africa’s *S. frugiperda* infestation may have originated in America [32,33]. In West Africa, two *S. frugiperda* strains have been identified, and both can attack maize crops. Other studies covering 22 sub-Saharan countries indicate a new *S. frugiperda* introduction in Africa, suggesting a second entry into Western Africa from a different source than the original [34]. These results underline the need for further research, including studies on other species of the same genus, to provide references for implementing sustainable and environmentally friendly *S. frugiperda* management methods to reduce maize production losses.

This study aims to provide a comprehensive review of the current knowledge on genetic characterization and pheromone-based mating disruption methods for the management of *Spodoptera* species, with a particular focus on *S. frugiperda*. Specifically, through a systematic approach, it seeks to compile information on genetic characterization, with an emphasis on the molecular markers used for accurate species and/or strain identification. Additionally, it aims to examine the advances in the implementation of the pheromone-based control method across *Spodoptera* species, including the types of synthetic sexual pheromones employed and their effectiveness. This study also explores the efficacy of the mating disruption method, addressing the challenges associated with establishing pheromone blends that account for strain diversity. Ultimately, this review identifies research gaps and highlights priority areas for further investigation to enhance the sustainable management of *S. frugiperda* in newly invaded regions.

## 2. Materials and Methods

### 2.1. Database Search Strategy

This systematic review was conducted in accordance with the PRISMA (Preferred Reporting Items for Systematic Reviews and Meta-Analyses) guidelines [35]. This review focused on publications related to the genetic characterization and use of sexual pheromones for mating disruption in *Spodoptera* species, particularly *S. frugiperda*, a global agricultural pest.

A search was conducted across three online databases: Scopus (https://www.scopus.com/search/form.uri?display=advanced (accessed on 19 September 2024)), Web of Science (https://www.webofscience.com/ (accessed on 15 September 2024)), and CABI Digital Library, a subject-specific database on agriculture relevant to the research topic under study (https://www.cabidigitallibrary.org/search/advanced/ (accessed on 13 September 2024)).

The search was comprehensive, without restrictions on the publication period, including articles published anytime up to September 2024. However, this review is limited to only articles in the English language, both full texts and abstracts, articles with the full text available, and publication types focusing on original research articles relevant to the review topics. Search queries were organized into three categories: (1) genetic characterization and strain identification in *Spodoptera* species; (2) sexual pheromones and mating disruption in *Spodoptera* species; and (3) the effectiveness of sexual pheromones for mating disruption in relation to genetic characterization and strain identification in *Spodoptera* species. For each category, basic and specific keywords were used, as listed in Table 1. Boolean operators (AND/OR) and truncation symbols were combined with these terms to formulate the search queries.

The search codes and results for each database are provided in Appendix A, and a table with all articles identified was published in the online repository Figshare [36] and all references managed in Mendeley software for desktop (v. 2.137.0; Elsevier, London, UK) [37].

### 2.2. Screening and Data Curation Workflow: Inclusion and Exclusion Criteria of Relevant Studies

During the literature search, the initial identification resulted in a significant number of articles from each database, which were subsequently refined by removing publications that were not journal articles, such as book chapters or proceedings. After records were assembled from the three databases, duplicate entries—defined as articles retrieved simultaneously from the different databases—were identified and removed using a Microsoft Excel tool (v.2403; Microsoft Corporation, Redmond, WA, USA) [38]. This step was essential to ensure that there was no duplication among databases and that each contributed with unique records not captured by the others. The overlap between databases was quantified, as illustrated in a Venn diagram (Appendix A).

Subsequently, the screening process conducted via manual curation was carried out for the assessment of title relevance and the content of both abstracts and full texts, applying inclusion and exclusion criteria. The inclusion criteria encompassed only original research articles related to genetic and pheromone-based control on *Spodoptera* species. Two exclusion criteria were subsequently applied. The first exclusion criterion involved the removal of articles that did not meet the topics under study, namely the following: (i) articles not related to *Spodoptera* species (e.g., other insect species or taxa); (ii) articles not relevant to the main topics of this review (e.g., biological control, insecticide resistance, or chemical management); and (iii) review articles. The second exclusion criterion was based on the assessment of the abstract and full-text content of each article by evaluating the relevance of the information to the topics under study. Thus, (i) articles with inaccessible full text or abstracts lacking sufficient data for analysis; (ii) articles with abstracts or full texts in languages whose data were not accessible; and (iii) other article types not identified during the first exclusion (e.g., reports, proceedings) were eliminated for subsequent data processing. For retrieved records, the full text was obtained directly from databases (i.e., open access articles) or alternative sources (e.g., publisher websites, b-on—https://www.b-on.pt (accessed on 22 November), institutional repositories, or directly from the corresponding author). For records indexed at CABI Abstracts, the full texts were obtained through CABI Library institutional voucher account access. After applying the two exclusion criteria, the remaining articles were considered eligible for inclusion in this review (Figure 1). The details of the eligible articles included in this systematic review are provided in Appendix A.

### 2.3. Data Extraction and Analysis

Data extraction was performed using a standardized Excel sheet to ensure consistency across studies. A subset of records was checked by two reviewers independently to verify the accuracy and standardization of the extracted data and validate reproducibility. The resulting table included general information such as the article title, authors, year of publication, country, region of study, *Spodoptera* species studied, and associated host crops.

For articles on genetic studies, details on the molecular methodology used, genetic markers employed, and hybrid detection capabilities were documented. For articles on pheromone studies, the table captured aspects including the study components covered (reproductive biology in the laboratory, mating disruption in the field, or both), specific active substances of the pheromones, and their concentration or proportion in each commercial blend. Additionally, the geographical origin of insects used for extraction, the dosage of pheromones (number of dispensers per hectare), and whether pheromones were applied according to strain were recorded. The mechanisms of mating disruption employed and the effectiveness of these pheromones, indicated by a disruption index (negative < 70% and positive > 70% reduction in capture), were also included. Finally, the effect of mating disruption on reducing the insect population (eggs, larvae, adults) was noted. All extracted information was organized in the comprehensive Appendix A.

The data collected were analyzed using various software, such as Microsoft Excel (v. 2403) and QGIS (v.3.28.4; QGIS Development Team, Zurich, Switzerland) [38,39]. Excel was used to create a Venn diagram illustrating the overlap between the three search topic categories in each database and to create a bar diagram illustrating the types of genetic markers. QGIS was utilized to create a global map illustrating the distribution of the selected scientific publications. The associated host crops, the geographical concentration of mating disruption studies for each *Spodoptera* species, and the main genetic markers used in the studies were presented through Sankey diagrams, produced using the RAWGraphs tool (v.2.0; DensityDesign Lab, Milan, Italy) [40].

## 3. Results

### 3.1. Characterization of Included Sources

#### 3.1.1. Number of Studies Identified and Included in This Review

A total of 4523 articles were retrieved from the three databases: 3255 from Scopus (72%), 687 from Web of Science (15%), and 581 from the CABI Digital Library (13%). Before starting the manual screening process, 1487 articles were excluded after applying the refined search filters in Excel, removing duplicates between the three search topic categories in each database, overlapping duplicates between databases, and articles classified as reviews. The overlap among the databases was as follows: 374 articles shared between Scopus and Web of Science, 340 between Scopus and CABI Digital Library, and 43 between Web of Science and CABI; 178 articles were common to all three databases (Appendix A). This process resulted in 3036 unique articles (2974 from Scopus, 457 from Web of Science, and 5 from CABI), after title screening. Of these, 2601 articles were excluded due to irrelevance to *Spodoptera* species or the review topics or because they were review articles. The remaining 435 articles underwent full-text or abstract review, with 328 excluded due to inaccessibility, language issues (non-English), or insufficient data.

In total, 107 articles met the inclusion criteria and were included in this review (Appendix A). Of these, 84 articles (78%) only focused on the genetic characterization of various *Spodoptera* species, with an emphasis on genetic diversity, population structure, and evolutionary relationships, while 16 articles (15%) investigated the use of sexual pheromones for mating disruption in *Spodoptera* species, evaluating the effectiveness of pheromone-based strategies for pest population control. Only seven articles (7%) addressed both genetic characterization and the use of sexual pheromones for mating disruption, exploring the intersection between genetic variability and the success of mating disruption techniques. The diagram in the Appendix A illustrates the distribution of articles by research topic addressed in this review.

#### 3.1.2. Geographical and Temporal Distribution of Studies

The temporal analysis of the articles, with publications ranging from 1976 to 2024, showed a marked increase in the number of publications beginning in 2010 (Appendix A). From 1976 to 2000, fewer than three articles were published per year, and the publication rate began to rise during the 2000s, increasing significantly with the invasion of *S. frugiperda* in Africa in 2016. This trend continued with a notable increase in the number of articles published after the detection of the pest in Asia in 2020. During this period (2000–2024), 63% of all articles were published, which coincides with the spread of *S. frugiperda* to new regions, particularly in Africa and Asia. The surge in publications after this period is largely attributed to the growing focus on the genetic identification, behaviour, and other biological aspects of the pest, particularly as it spread to new regions, prompting increased research on its population dynamics and control strategies.

The geographical focus of the reviewed articles covers various continents (Figure 2), with most studies including data from the Americas (48 articles, 42%), followed by Asia (38 articles, 34%), Africa (22 articles, 19%), Europe (3 articles, 3%), and Oceania (1 article, 1%). In the Americas, 94% of the articles focused on *S. frugiperda*; in Africa, about 98% of the articles focused on this species; and in Oceania, all articles were dedicated to this species. In Asia, *S. frugiperda* was studied in 69% of the articles, while *S. litura* and *S. exigua* comprised 17% and 10%, respectively. In Europe, all articles focused exclusively on *S. littoralis*.

On the number of studies on *S. frugiperda*, the United States of America leads the research effort in the Americas, with 18 articles, followed by Brazil (12), Argentina (7), and Colombia and Mexico (6 each) (Figure 2). In Asia, *S. frugiperda* research is concentrated in India (seven) and China (five). In Africa, significant contributions have come from Kenya (seven), Togo (six), and Ghana and Tanzania (four each). Most articles in Africa, particularly in West Africa, have focused on genetic characterization. This distribution underscores the global research effort and highlights the agricultural and invasive significance of *S. frugiperda*, especially in newly invaded regions, such as West Africa and Asia.

#### 3.1.3. Spodoptera Species and Associated Host Crops

The analysis of the articles revealed a strong focus on several *Spodoptera* species, each associated with specific host crops. *S. frugiperda* was the most extensively studied species, with 90 articles dedicated to confirming its identification and understanding its biology, behaviour, and management strategies. Most of these records (84 articles) focused on the genetic characterization of the species. Studies conducted on *S. frugiperda* were the most associated with host crops such as maize, rice, cotton, and sorghum (Figure 3). Other species investigated included *S. exigua* (6 articles), with cotton and onion as primary host crops, and *S. litura* and *S. littoralis*, studied in 10 and 4 articles, respectively. Less commonly studied species included *S. eridania* and *S. cosmioides*, each the subject of a single article. The 13 articles on mating disruption methods for *S. frugiperda* were associated with crops such as maize, rice, sorghum, grasses, and cotton. Similarly, *S. exigua* and *S. cosmioides* were also associated with cotton. Thus, only these two species were studied on the same host crop as *S. frugiperda*, i.e., cotton.

### 3.2. Data Analysis on Genetic Characterization and Pheromone-Based Mating Disruption in Spodoptera Species

#### 3.2.1. Genetic Characterization and Molecular Methodology Employed

During the process of data analysis and synthesis, a crucial aspect examined consisted of the advancements driving genetic research, particularly focusing on the genetic markers used for strain and hybrid identification. A Sankey diagram (Figure 4) presents the distribution of the genetic markers across the studies screened, offering a clear representation of prevailing trends in marker usage within genetic research on each *Spodoptera* species.

Upon comprehensive analysis, a significant majority of genetic studies (76% of the articles on genetic characterization of *Spodoptera* species) included markers categorized as “Barcoding,” with the *COI* (Cytochrome c oxidase subunit I) gene being the predominant marker. Notably, 32% of the genetic characterization articles incorporated markers in the “Genomic” category, with the *Tpi* (Triosephosphate Isomerase) gene being a key contributor, while about 20% employed diverse markers categorized as “Population Genetics,” including SNPs (Single-Nucleotide Polymorphisms), Microsatellites (or SSRs—Simple Sequence Repeats), AFLPs (Amplified Fragment Length Polymorphisms), and RAPD (Random Amplified Polymorphic DNA). Across all *Spodoptera* species, the most used genetic marker for species identification is the *COI* gene (used in 76% of the genetic characterization articles). In the context of *S. frugiperda*, which is predominantly studied for genetic characterization, both the *COI* and *Tpi* genes are frequently employed, often in combination (29% of the articles). However, various other markers have also been utilized for this species, including Microsatellites, SNPs, and AFLPs, among others, especially for assessing population structure within *Spodoptera* species. Recent studies have highlighted the need for a more comprehensive understanding of genetic diversity, particularly in newly invaded regions, such as West Africa, where the research focus is intensifying, especially in aspects of genetic characterization.

The analyzed data also shows a variety of molecular methodologies and types of genetic markers employed to detect hybrids, specifically in *S. frugiperda* (Appendix A). Among the articles reviewed that focused on hybrid detection, positive results were only reported in 23 articles. Among the molecular methodologies utilized, “Barcoding” was the most common, with the *COI* genetic marker alone yielding positive results in only 3 studies detecting hybrids, while 41 studies did not detect any hybrids. The combination of different molecular methodologies and types of genetic markers showed varying outcomes. For instance, combining the *COI* (Barcoding) and *Tpi* (Genomic) markers resulted in 10 positive responses for hybrid detection, while an equal number of 10 studies did not detect hybrids. Conversely, other molecular methodologies classified under “Population Genetics” did not result in positive findings for hybrid detection; for example, four studies using AFLPs reported no hybrids detected. Additionally, the use of Microsatellites alone led to two positive detections and one non-detection, while other markers, such as RAPD and SNPs, showed no positive results for hybrid detection. These findings underscore the diversity of molecular approaches employed in hybrid detection in *S. frugiperda*, as well as the variation in outcomes obtained with different combinations of methodologies and genetic markers. Overall, the analysis reflects a complex landscape of hybrid detection, highlighting the importance of selecting appropriate molecular techniques to enhance the understanding of genetic diversity within these species.

#### 3.2.2. Pheromone-Based Mating Disruption Strategies in *Spodoptera* Species

By analyzing the geographical distribution of studies conducted on mating disruption methods (Figure 5), this review found few studies on *S. frugiperda*, predominantly conducted in America (nine studies), followed by Africa (three studies) and Asia (two studies). All research focused on *S. litura* was conducted in Asia, while studies on *S. littoralis* were conducted exclusively in Europe. *S. exigua* was the subject of three studies, two conducted in Asia and one in America.

The implementation of mating disruption in the field was achieved in various *Spodoptera* species. A review of pheromone-based mating disruption studies conducted on *Spodoptera* species reveals predominant research in laboratory settings and on *S. frugiperda*, which has been extensively studied. The results showed that there were nine laboratory studies mainly conducted on maize [23,41,42,43,44,45,46,47,48], only one study conducted on the field [49], and three studies combining both the laboratory and field, two of which were on maize [50,51] and one on cotton [52]. Other species such as *S. litura*, *S. exigua*, *S. littoralis*, and *S. cosmioides* had limited research conducted on them, primarily in field conditions, for instance, *S. exigua* on cotton [53] and onion [54,55] and *S. litorallis* in spinach [56]. These findings highlight a predominant focus on *S. frugiperda*, particularly in laboratory settings, with fewer studies extending to field conditions or combining both approaches.

Regarding the type of synthetic sexual pheromone employed, this review reveals that a range of single-component formulations and various blended formulations were used in mating disruption studies for different *Spodoptera* species (Table 2). For *S. frugiperda*, Z9-tetradecenyl acetate (Z9-14:Ac) was the most employed pheromone, either singly or in combination with other components like Z9-dodecenyl acetate (Z9-12:Ac), Z7-dodecenyl acetate (Z7-12:Ac), and Z11-hexadecenyl acetate (Z11-16:Ac). Also, this review reveals that Z9-14:Ac is the most frequently observed component in the pheromone blends employed across various *Spodoptera* species. While most studies demonstrated the positive effectiveness of these pheromone blends in mating disruption, specific adult capture rates were often not reported. However, a recent study on *S. frugiperda* employed a blend combining Z9-14:Ac (77.49%) and Z11-16:Ac (11.58%) and achieved an 87% capture rate [49]. The pheromone blends used in *S. frugiperda* specifically for corn and rice strains were also effective, but details on the capture rate were not provided. Studies conducted on *S. exigua* reported capture rates between 93% and 97.8% [53,54], and on *S. littoralis*, 98.9% was achieved [56]. In contrast, the effectiveness on *S. cosmioides* [53] and *S. litura* was usually not detailed [57,58,59].

#### 3.2.3. Effectiveness and Impact of Mating Disruption Method on *Spodoptera* Species

Few studies provide comprehensive details on pheromone concentrations or dispenser densities, both of which are crucial factors influencing the effectiveness of mating disruption methods. The studies that provide this information reveal considerable variation in the methods used. For example, in field studies conducted on *S. exigua*, a high number of dispensers were applied, including 1000 dispensers per hectare in onion fields and 500 dispensers per hectare in cotton fields [53,54,55]. These studies used a blend of Z,E-9,12:Ac and Z9–14:OH, with the total pheromone dose varying by study. One study used 500 dispensers per hectare, delivering a total of 160 g of pheromone per hectare, which resulted in a 95% reduction in larval populations [53]. On the other hand, a more recent study on *S. frugiperda* applied fewer dispensers (40–60 per hectare) for mating disruption using a blend of Z9-14:Ac (77.49%) and Z11-16:Ac (11.58%) [49]. In this case, each dispenser delivered about 2.8 g of pheromones, totalling 112–168 g per hectare, nearly nine times more than in the study on *S. exigua* [53]. This study on *S. frugiperda*, which applied a higher pheromone load per dispenser, achieved up to 87% suppression in male trap captures, demonstrating the effectiveness of mating disruption [49]. Despite reductions in adult male populations, no significant changes in crop yield were observed, possibly due to the migration of gravid females from untreated neighbouring fields [54]. Earlier studies reported up to 80% reduction in adult *S. frugiperda* populations with specific pheromone blends in maize and grass crops, although impacts on crop damage were not always directly linked to population reductions [53]. In contrast, a study on *S. exigua* in onion fields in Japan observed reductions of 6% in eggs and 4% in larvae, linking pest suppression to potential crop loss reduction [54], although the level of reduction in crop loss was not reported.

Overall, these studies emphasize the importance of both pheromone dose and dispenser density, but they also reveal the challenges in translating population reductions into measurable reductions in crop damage. Further studies are needed to refine dispenser densities, pheromone compositions, and application methods to optimize mating disruption strategies across various *Spodoptera* species and field conditions.

#### 3.2.4. Impact of Genetic Variability, Pheromone Geographical Origin, and Composition on Mating Disruption Effectiveness

Most existing studies originate from the Americas, the pest’s native range, while relatively few have been conducted in regions where the pest has been recently reported, for instance, in Europe. This imbalance in research efforts poses a major limitation for making direct comparisons of its impacts across regions overall. Of the articles reviewed, most studies showed that limited attention has been given to genetic variability across *Spodoptera* species in mating disruption assessments. None of the studies specified the origin of insects used for pheromone formulation, although the manufacturer or supplier was usually mentioned, mostly from the same region where the study was conducted (America, Asia, or Europe).

This review also examined the proportion of “positive” and “negative” outcomes regarding mating disruption effectiveness according to strains, including geographic variation in pheromone responses and mating between strains. In terms of genetic variability, only one study on *S. frugiperda* considered the use of pheromone type according to the specific strain, and it was demonstrated that this approach led to an 80% reduction in adults, suggesting that strain-specific pheromone blends may enhance male suppression [25].

Regarding geographic variation in pheromone responses, one study on *S. frugiperda* populations in maize fields in Mexico showed no difference in effectiveness between blends from local and non-local populations [23]. Also, another study on corn- and rice-strain males demonstrated regional differences: corn-strain males varied in attraction to Z7-12:Ac, while rice-strain males responded more uniformly [51]. This study showed that adding Z11-16:Ac did not improve attraction in Florida or Peru, whereas supplementing Z9-14:Ac with its stereoisomer E7-12:Ac increased captures in North Carolina but not in Peru. These results demonstrate that both strain identity and regional differences in pheromone response influence male captures and should be considered when optimizing mating disruption strategies. This underscores the need for more comprehensive and field-validated research to develop effective mating disruption strategies tailored to different *Spodoptera* species and their host crops. Standardized methodologies and integrated research efforts are essential to address the complex interactions between genetic variability, pheromone-based control methods, and pest management strategies.

## 4. Discussion

This review provides insights into various aspects of *Spodoptera* species research, focusing on genetic characterization, mating disruption control methods, and their application for *S. frugiperda*. By analyzing the results of several studies, this review highlights general trends, regional differences, and gaps in research that inform integrated pest management. Understanding these patterns requires considering the global distribution of *Spodoptera* species and proper species and strain identification at the local level.

Among *Spodoptera* species, *S. frugiperda* has emerged as the most studied species worldwide due to its global distribution and significant impact on economically important crops like maize, rice, and cotton. First reported as a major pest in the Americas in the mid-19th century, its spread to Africa in 2016, and later to Asia in 2020, has raised concerns over its potential to cause greater damage to maize than other *Spodoptera* species in these regions [61,62]. This has triggered a surge in research, particularly in Africa and Asia. While Africa has seen numerous studies focusing on species identification and behavioural understanding, especially in East Africa, with countries like Tanzania and Kenya leading these efforts, Asia, despite the relatively recent introduction and spread of *S. frugiperda*, has already produced a few but significant studies addressing its impact and management. This highlights the importance of continued research in these regions to develop effective pest management strategies. In particular, the need for more balanced research across the African continent remains critical, as West Africa, the original entry point for *S. frugiperda*, is making the least progress in scientific efforts.

### 4.1. Genetic Characterization of Spodoptera Frugiperda: Which Approach Should Be Used for Accurate Identification?

Genetic characterization is critical for species and strain identification. Many studies have focused on confirming species identification using genetic markers, with the *COI* and *Tpi* genes being the most used. The *COI* gene, a mitochondrial marker, is highly effective for species identification due to its high mutation rate and maternal inheritance, but it is less reliable for detecting intraspecific variation [62,63]. In contrast, the *Tpi* gene, a nuclear marker with biparental inheritance, provides a more stable evolutionary rate and is better suited for strain identification [28]. For *S. frugiperda*, distinguishing between the two strains (corn and rice) is essential, as they differ in mating behaviour and host plant preference [63]. To date, the results highlight that both the *COI* and *Tpi* genes are valuable for species identification, but controversy remains regarding the distinction between rice and corn strains, as the results vary depending on the genetic markers analyzed [62]. Combining both markers can enhance accuracy in strain identification and help detect hybridization within populations. Research has revealed significant genetic differences between populations in East and West Africa, with the low prevalence of the rice strain even in rice habitats [64] further complicating strain identification.

These findings highlight the importance of carefully selecting genetic markers for accurate strain identification in research and pest management strategies.

### 4.2. Pheromone-Based Control Strategies: Advances and Challenges for Mating Disruption Effectiveness for Spodoptera frugiperda

Pheromone-based mating disruption has shown promise for controlling *Spodoptera* species, particularly *S. exigua* and *S. littoralis* [53,55,56]. However, its application to *S. frugiperda* has been less consistent across regions, reflecting variability in pest presence or distribution and in pheromone blend composition, environmental conditions, and pest population dynamics.

This review found that various sexual pheromone components, both singly and in mixtures, have been used in attempts to control this pest. While laboratory studies have demonstrated population reductions using pheromone blends [41], field trials have yielded mixed results. The variability in effectiveness is influenced by factors such as pheromone blend composition, environmental conditions, and pest population dynamics. A major challenge remains the lack of standardized pheromone blends specifically tailored to *S. frugiperda*.

The sexual pheromone of *S. frugiperda* was first identified nearly four decades ago, revealing two main components: Z-9-14:Ac and Z-7-12:Ac [50]. In fact, this review found that these compounds remain the most used in pheromone-based strategies, either individually or in combination. However, Z-9-14:Ac is shared by several *Spodoptera* species, reducing specificity and increasing the risk of non-target captures [45]. It was demonstrated that commercial blends with three or four pheromone components for *S. frugiperda* had a high non-target capture rate specifically of *Leucania phragmatidicola* (Guenée, 1852), which compromised monitoring efforts in the northeastern United States [65]. Such findings highlight the importance of the region-specific validation of pheromone blends, as *S. frugiperda* populations in West Africa, mainly corn-strain and rice–corn hybrids, exhibited higher proportions of Z-7-12:Ac and increased male sensitivity to this component compared to American populations [43]. This shift may serve to reduce interference from other local *Spodoptera* species sharing the major component, emphasizing the need to adapt sexual pheromone formulations to local populations for effective monitoring and/or for control by mating disruption. These observations suggest that the effectiveness of any pheromone trapping and/or control method can be influenced by the genetic population composition in a specific region, underscoring the need for assessment rather than assuming broad applicability.

Additionally, the geographic variation in *S. frugiperda* males’ response to specific minor pheromone components, such as Z7-12:Ac, Z11-16:Ac, and the isomer E7-12:Ac, alongside the major component Z9-14:Ac and pheromone lure composition, highlights the need for further studies in specific regions and contexts to better understand strain identity and regional pheromone composition, optimizing mating disruption [23,51].

Optimizing these components is essential for effective monitoring and control. Indeed, field trials have shown that slight adjustments in the ratios of these components can reduce non-target captures while maintaining high efficacy in trapping *S. frugiperda* [45,66]. A recent study also highlighted that the inclusion of minor components like nonanal, an aldehyde commonly used in fragrances, can optimize trap specificity for *S. frugiperda* [67]. This modification improved the attraction of male moths while effectively reducing the capture of non-target species, thereby enhancing the overall efficacy of pheromone-based monitoring and control strategies. This study emphasized that adjusting the composition of the sexual pheromone components with minor components can help refine trap and/or control targeting, making it more selective and effective for pest management programs.

### 4.3. Integration of Genetic Studies with Pheromone-Based Approaches: Challenges for Spodoptera frugiperda Management

Integrating genetic characterization with pheromone-based strategies could enhance the effectiveness of mating disruption for *S. frugiperda*. Studies revealed that differences in mating behaviour between the two strains of *S. frugiperda*, rice and corn, are genetically driven, with rice-strain females producing higher amounts of Z-7-12:Ac and Z-9-12:Ac compared to corn-strain females [25]. This strain-specific variability suggests that tailored sexual pheromone blends, based on genetic differences, could improve the success of mating disruption.

However, studies from West Africa, including Benin and Nigeria, found no significant strain-specific differences in pheromone composition or responses, despite the presence of two distinct strains [43], suggesting regional adaptations that could influence the success of pheromone-based control methods in Africa, where populations differ from those in the Americas. The absence of standardized sexual pheromone blends for specific strains or regions remains a significant challenge for sustainable *S. frugiperda* management, especially in newly invaded areas. The genetic characterization of agricultural pest populations, such as *S. frugiperda*, plays a crucial role in understanding their population structure, dispersal patterns, and adaptation to different agroecosystems. This knowledge is fundamental for guiding specific pest management strategies, including the use of semiochemicals, such as sexual pheromones, according to species or strains identified by molecular approaches. The application of sexual pheromones, whether for monitoring or for control proposal such as mating disruption, offers an environmentally sustainable alternative to chemical insecticides, reducing their negative impact on non-target species, human health, and ecosystems.

The integration of genetic data with the mating disruption method to reduce the population level enhances effectiveness by providing insights into population structure, potential genetic variation, and behavioural responses to pheromone cues. The incorporation of genetic insights and pheromone-based strategies into integrated pest management programs will enhance the accuracy and efficacy of pest control, aligning biotechnological and ecological principles to better strengthen crop protection against invasive species.

### 4.4. Gaps and Future Directions in Mating Disruption Research for Spodoptera frugiperda

The analysis of the existing literature on mating disruption for *S. frugiperda* reveals several important gaps, particularly concerning the optimization of sexual pheromone concentrations, dispenser densities, and application methods. While some studies have provided valuable insights into the efficacy of different dispenser densities [49,53], others often fail to provide sufficient detail on the dosage of the various components of the synthesized pheromone mixture used or the specific mechanisms by which mating disruption is achieved. The absence of standardized reporting for essential parameters, such as dispenser density and the doses of the racemic mixture used, limits the establishment of conclusive insights into the most effective application methodology.

The differences in effectiveness between studies, like those observed among *S. exigua* and *S. frugiperda*, underscore the complexity of achieving successful mating disruption. Although high dispenser densities are commonly used in studies on other *Spodoptera* species, research on *S. frugiperda* suggests that even relatively low densities can cause partial mating disruption. This indicates that factors beyond dispenser density—such as pheromone dose and behavioural responses—also play a critical role in determining the success of the method. Additionally, there is a noticeable lack of consistent and standardized crop damage and yield assessments in many mating disruption studies. Although some studies report reductions in pest populations, translated into tangible benefits, these have not reported the impact on decreased crop damage or increased yields are missing. Without quantitative data on the impact of mating disruption on crop yields, the practical applications of this method remain unclear. Furthermore, the focus on trap catch reduction as a proxy for success may not accurately reflect the method’s overall effectiveness in real-world agricultural settings. To improve the practical relevance of mating disruption, future studies should prioritize crop yield and damage assessments alongside traditional population monitoring.

The effectiveness of mating disruption may also be influenced by landscape factors and field size, as suggested by the issue of migration (mainly of fertilized females) from untreated neighbouring fields. In smallholder farming systems, where fields are often smaller and more fragmented, the potential for pest immigration from untreated areas could undermine the effectiveness of mating disruption. This calls for a further exploration of how to optimize pheromone distribution and dispenser placement in such contexts. Moreover, integrating mating disruption with other control methods, such as biological control or push and pull, could enhance its overall efficacy and provide a more holistic approach to integrated pest management [49].

Finally, addressing the ecological factors that affect the dispersal of sexual pheromones is an area that is still in need of more attention. In larger, more homogenous fields, pheromone dispersion may be more even, while in smaller, fragmented fields, edge effects could significantly impact the success of mating disruption. Research should explore how to minimize these edge effects and optimize the placement of dispensers to ensure consistent pheromone exposure throughout the target area. Additionally, tailoring pheromone blends to local pest populations and environmental conditions may further improve the effectiveness of mating disruption. These research gaps underscore the need for a more comprehensive and standardized framework to evaluate mating disruption. Future work should emphasize comparative trials across different ecological contexts, consider integration with other pest management methods, and adopt standardized protocols for assessing crop-level outcomes. Ultimately, optimizing pheromone-based mating disruption for *S. frugiperda* requires integrating knowledge of the pest’s biology and strain variation with local agricultural practices and environmental conditions to maximize efficacy and crop protection.

## 5. Conclusions

Understanding the dispersal behaviour, reproductive biology, and population structure of *S. frugiperda* is essential to ensure effective population reduction. Incorporating molecular tools to identify genetic variation and strain composition across different agroecological contexts could support more precise pest management efforts. Pheromone-based mating disruption adapted to the specific agroecological context could enhance integrated pest management programs by disrupting the pest’s reproduction, reducing population pressure over time, and enabling compatibility with other control methods such as biological control and chemical control (preferably with selective insecticides), thereby minimizing environmental impact and pest resistance to pesticides.

The pest’s migratory behaviour and the potential introduction of new invasive lineages or hybridization highlight the importance of continuous genetic monitoring and locally adapted management strategies. Integrating population genetic studies with semiochemicals, particularly the most effective pheromone blends, offers a promising approach to sustainable pest control. Further research should focus on the following: (1) mapping pheromone components and proportion with their genetic determinants across populations, validating their effects on behavioural responses; (2) characterizing pheromone production and response in local populations; and (3) developing pheromone-based mating disruption methods tailored to local populations, potentially combined with host plant resistance and biological control to enhance integrated pest management.

In conclusion, this research underscores the relevance for a comprehensive and interdisciplinary approach to the management of *S. frugiperda*, particularly in sub-Saharan Africa, where its rapid spread is significantly impacting food security. Accurate genetic characterization is essential for strain identification and understanding population dynamics, which in turn supports effective control strategies.

## Figures and Tables

**Figure 1 insects-16-01176-f001:**
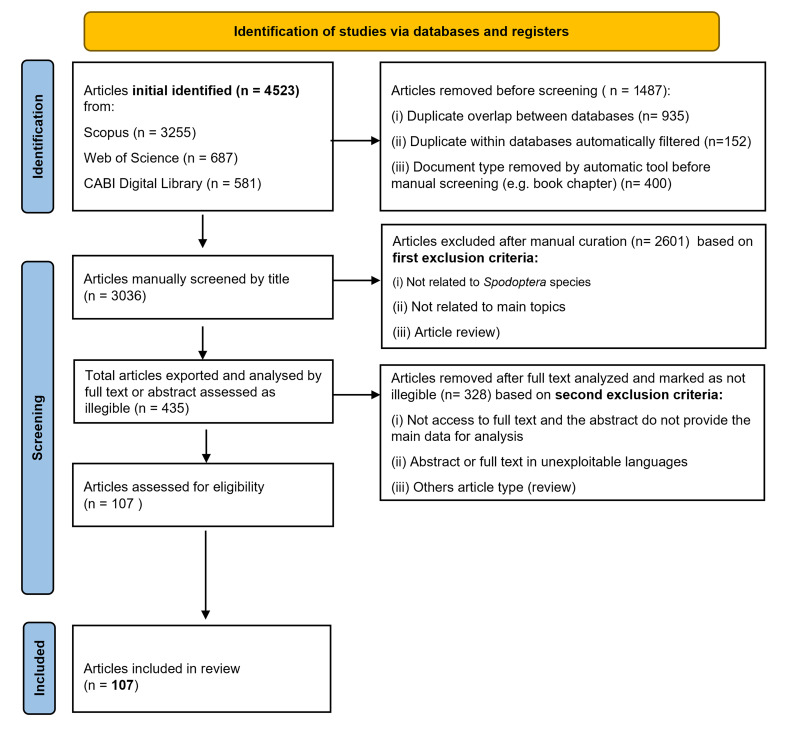
A PRISMA 2020 flow diagram for the selection of articles included in this systematic review on genetic characterization and pheromone-based strategies for mating disruption in *Spodoptera* species. The numbers represent the articles recorded at each stage of the selection process: identification (total number of articles initially identified in each database), screening (number of articles selected after removal of duplicates and by manual curation according to the first exclusion and second exclusion criteria and included in final review analysis (number of final articles included in this review).

**Figure 2 insects-16-01176-f002:**
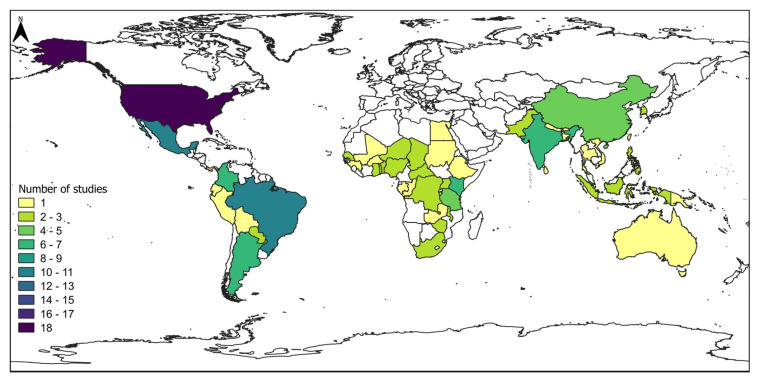
Geographical distribution of articles on *S. frugiperda* by country.

**Figure 3 insects-16-01176-f003:**
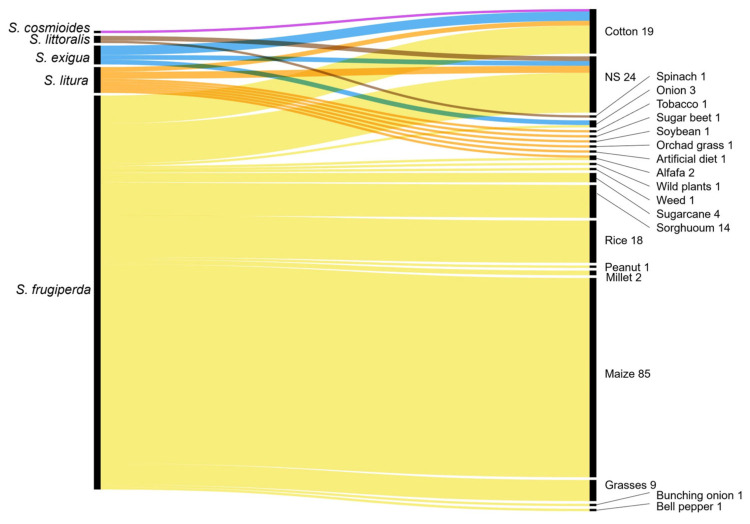
A Sankey diagram illustrating the *Spodoptera* species studied and their associated host crops.

**Figure 4 insects-16-01176-f004:**
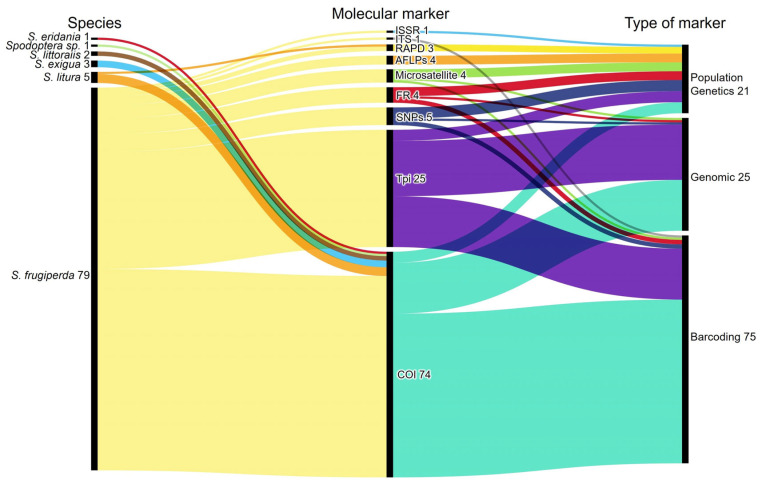
This Sankey diagram illustrates the main types of genetic markers used for genetic characterization and the number of records for each *Spodoptera* species. Types of genetic markers: Barcoding—*COI* (Cytochrome c oxidase subunit I) and ITS (Ribosomal Internal Transcribed Spacer); genomic markers—*Tpi* (Triosephosphate Isomerase) and FR (For Rice). Population Genetics—SNPs (Single-Nucleotide Polymorphisms), Microsatellites, AFLPs (Amplified Fragment Length Polymorphisms), RAPD (Random Amplified Polymorphic DNA), and ISSR (Inter-Simple Sequence Repeat of DNA).

**Figure 5 insects-16-01176-f005:**
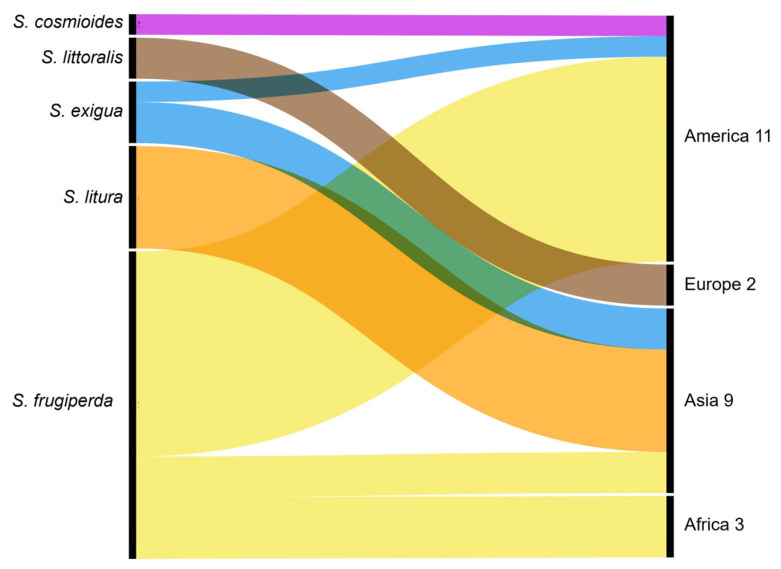
A Sankey diagram representing the geographical distribution of studies conducted on each *Spodoptera* species with the mating disruption method: *S. frugiperda*, *S. litura*, *S. exigua*, *S. littoralis*, and *S. cosmioides*.

**Table 1 insects-16-01176-t001:** Keywords used for search categories, including basic search keywords common to all main topics and topic-specific keywords. (* sexual.)

Main Search Topic	Basic Search Keywords	Topic-Specific Search Keywords
(1) Genetic characterization and strain identification in *Spodoptera* species	“*Spodoptera*”/ “*Spodoptera frugiperda*”/“fall armyworm”	“genetic characterization” /“molecular identification” /“strain identification” /“genetic marker”
(2) Sexual pheromones and mating disruption in *Spodoptera* species		“sex * pheromone”/“mating disruption”
(3) The effectiveness of sexual pheromones for mating disruption concerning genetic characterization and strain identification in *Spodoptera* species		“strain identification” /“genetic characterization” /“sex * pheromone”/“mating disruption”

**Table 2 insects-16-01176-t002:** Type of sexual pheromone employed by *Spodoptera* species, effectiveness in mating disruption control (Yes—Positive; No—Negative; NS—Not Specified), and adult capture rates (%) in reviewed studies.

Spodoptera Species	Pheromone Type/ Active Components (Proportion)	Effectiveness/Capture Rate (%)	References
*S. frugiperda*	Single: Z9-14:Ac/Z9-12:Ac	Yes/NS	[23,43,45,50]
	Single: Z11-16:Ac/Z7-12:Ac/E7-12:Ac	Yes/NS	[23,43,45,46,47]
	Blend: Z9-14:Ac + Z9-12:Ac (10:1)	N/S/NS	[52]
	Blend: Z9-14:Ac (77.49%) + Z11-16:Ac (11.58%)	Yes/87%	[49]
	Blend: Z9-14:Ac (87%) + Z11-16:Ac (12.5%) + Z7-12:Ac (0.5%)	NS/NS	[49]
	Blend: Z9-14:Ac (30%) + Z7-12:Ac	Yes/NS	[41]
	Blend: Z9-14:Ac (30%) + Z11-16:Ac + Z7-12:Ac	Yes/NS	[41]
	Blend: Z9-14:Ac (90%) + Z7-12:Ac + E7-12:Ac + Z9–12:Ac + Z11–16:Ac	NS/NS	[41]
	Corn blend: Z9-12:Ac (1%) + Z7-12:Ac (2%) + Z11-16:Ac (13%); Rice blend: Z9-12:Ac (2%) + Z7-12:Ac (4%) + Z11-16:Ac (8%)	Yes/80%	[25]
*S. exigua*	Blend: Z9-12-14:Ac + Z9-14:OH (7:3)	Yes/93–97.8%	[53,54,55]
*S. littoralis*	Blend: Z9-11:Ac + Z9-12:Ac (95:5)	Yes/98.9%	[56]
*S. cosmioides*	Blend: Z9-14:Ac + Z9-12:Ac (10:1)	NS/NS	[53]
*S. litura*	Single: Z9-11:Ac; Single: Z9-12-14:Ac	NS/NS	[57,58,59]
	Single: Z9-14:OH	No/NS	[60]

## Data Availability

Raw data is available at the online repository Figshare (Dataset, https://doi.org/10.6084/m9.figshare.28135550.v1, accessed on 18 December2024), reference number [36]. All remaining data is available in the Appendix A.

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
