# Peer review of "Genetic Characterization and Mating Disruption in Spodoptera Species, a Case Study on Spodoptera frugiperda (Lepidoptera, Noctuidae): A Systematic Review"

_insects, 2025, doi:10.3390/insects16111176_

Round 1
Reviewer 1 Report
Comments and Suggestions for Authors
The manuscript entitled "Genetic characterization and mating disruption in Spodoptera species focusing on Spodoptera frugiperda: a systematic review" represents a review paper, or better called meta-analysis, with the focus on literature review of genetic strains of Spodoptera species, mating disruption and geographical extant of Spodoptera species. In general, the manuscript is well-written, with some minor comments. Also, this is a meta-analysis, not a review.
Genetic characterization and mating disruption in Spodoptera species focusing on Spodoptera frugiperda: a systematic review
Proposition for the title: Genetic characterization and mating disruption in Spodoptera species, focusing on Spodoptera frugiperda (Lepidoptera: Noctuidae): a systematic review
Simple Summary: This chapter is well-written, with only minor adjustments.
Line 19: Genus Spodoptera includes several species whose caterpillars cause...
Lines 19-20: Spodoptera frugiperda (J. E. Smith, 1797)
Line 21: pests’s
Abstract: This chapter is well-written, with only minor adjustments.
Line 30: Spodoptera frugiperda (J. E. Smith, 1797)
Line 34: local or different environmental conditions.
- Introduction: This chapter is well-written, with only minor adjustments.
Lines 48-50: Spodoptera frugiperda (J. E. Smith, 1797), S. exigua (Hübner, 1808), S. exempta (Walker, 1857), S. litura (Fabricius, 1775), and S. littoralis (Boisduval, 1833).
Line 53: ...specifically Nigeria and São Tomé and Príncipe,...
Line 62: You mention the term Fall armyworm for the first time here. Please, use the Spodoptera frugiperda or S. frugiperda throughout the text, so that you do not make a misunderstanding.
Lines 74, 78, 79, 91-92, 111, 113, 115-116: change fall armyworm to S. frugiperda
- Materials and Methods: This chapter is well-written.
- Results: This chapter is well-written, with only minor adjustments.
Lines 250, 252, 260, 261, 264, 266, 270, 278, 281, 286, 289, 318, 327, 340, 347, 358, 364, 369, 376, 378: S. frugiperda in italic, i.e., S. frugiperda
Line 262: S. litura and S. exigua in italic, i.e., S. litura and S. exigua
Line 263, 363-364, 380: S. littoralis in italic, i.e., S. littoralis
Lines 276, 277, 299, 309, 316, 322, 345, 356, 357, 369, 374, 385: Spodoptera in italic
Lines 283, 287, 350, 362, 363, 379: S. exigua in italic
Lines 284, 349, 362, 381: S. litura and S. littoralis in italic, i.e., S. litura and S. littoralis
Lines 285, 287, 362, 381: S. eridania and S. cosmioides in italic, i.e., S. eridania and S. cosmioides
- Discussion: This chapter is well-written, with only minor adjustments.
Line 537: delete fall armyworm
Line 547: change fall armyworm to S. frugiperda
Line 563-564: S. exigua in italic
Author Response
Alterations are highlighted with track changes in the revised manuscript, along with a clean version of the revised manuscript, and point-by-point responses to all comments are provided below. The comments made by the Reviewers are in bold font and our responses are in normal font.
Comments and Suggestions for Authors
The manuscript entitled " Genetic characterization and mating disruption in Spodoptera species focusing on Spodoptera frugiperda: a systematic review " represents a review paper, or better called meta-analysis, with the focus on literature review of genetic strains of Spodoptera species, mating disruption and geographical extant of Spodoptera species. In general, the manuscript is well-written, with some minor comments. Also, this is a meta-analysis, not a review.
Genetic characterization and mating disruption in Spodoptera species focusing on Spodoptera frugiperda: a systematic review
Q1. Proposition for the title: Genetic characterization and mating disruption in Spodoptera species, focusing on Spodoptera frugiperda (Lepidoptera: Noctuidae): a systematic review
R1. We thank the reviewer for the suggestion to change the title, which we also agree, as it better reflects the content of this study. Considering the recommendation from both reviewers 1 and 2, the title was changed accordingly to: “Genetic characterization and mating disruption, a case study on Spodoptera frugiperda (Lepidoptera: Noctuidae): a systematic review”
Simple Summary: This chapter is well-written, with only minor adjustments.
Q2. Line 19: Genus Spodoptera includes several species whose caterpillars cause...
R2. Considering the reviewer comment, the text has been improved in the simple summary section, L20 to “Spodoptera species includes several species whose caterpillars cause major crop losses, notably (…)”
Q3. Lines 19-20: Spodoptera frugiperda (J. E. Smith, 1797)
R3. As per reviewer suggestion, the scientific name has been updated at L21 “Spodoptera frugiperda (J. E. Smith, 1797), “.
Q4. Line 21: pests’s
R4. Corrected as recommended at L22.
Abstract: This chapter is well-written, with only minor adjustments.
Q5. Line 30: Spodoptera frugiperda (J. E. Smith, 1797)
R5. As per reviewer suggestion, the scientific name has been updated int the Abstract section, L33 “Spodoptera frugiperda (J. E. Smith, 1797)“.
Q6. Line 34: local or different environmental conditions.
R6. Modified as recommended for “different environmental conditions” both at the Simple Summary section at L23-24 “(…) but their success depends on understanding the pest’s biology, ecology and behavior in the different environmental conditions.”, and at the Abstract section, L36-L37: “requires knowledge of the pest in different environmental conditions and evaluation of pheromone effectiveness under agroecological conditions.”.
- Introduction: This chapter is well-written, with only minor adjustments.
Q7. Lines 48-50: Spodoptera frugiperda (J. E. Smith, 1797), S. exigua (Hübner, 1808), S. exempta(Walker, 1857), S. litura (Fabricius, 1775), and S. littoralis (Boisduval, 1833).
R7. As per reviewer comment, scientific names have been completed as recommended. In the revised manuscript, Introduction section can be read as follows at L65-L68 “The genus Spodoptera (Lepidoptera: Noctuidae) includes various species of severe agricultural pests found worldwide [1,2]. Notable species include the Spodoptera frugiperda(J.E. Smith, 1797), Spodoptera exigua (Hübner, 1808), Spodoptera exempta (Walker, 1857), Spodoptera litura(Fabricius, 1775), and Spodoptera littoralis (Boisduval, 1833).”
Q8. Line 53: ...specifically Nigeria and São Tomé and Príncipe,...
R8. Word “in” eliminated as recommended at L713.
Q9. Line 62: You mention the term Fall armyworm for the first time here. Please, use the Spodoptera frugiperda or S. frugiperda throughout the text, so that you do not make a misunderstanding. Lines 74, 78, 79, 91-92, 111, 113, 115-116: change fall armyworm to S. frugiperda
R9. The authors acknowledge the reviewer comment, thus in the revised manuscript, the text has been extensively reviewed for replacing the term fall armyworm with Spodoptera frugiperda or S. frugiperda. E.g., Introduction section at L80, L92, and from then onwards, as per reviewer suggestion.
- Materials and Methods: This chapter is well-written.
- Results: This chapter is well-written, with only minor adjustments.
Q10. Lines 250, 252, 260, 261, 264, 266, 270, 278, 281, 286, 289, 318, 327, 340, 347, 358, 364, 369, 376, 378: S. frugiperda in italic, i.e., S. frugiperda
Line 262: S. litura and S. exigua in italic, i.e., S. litura and S. exigua
Line 263, 363-364, 380: S. littoralis in italic, i.e., S. littoralis
Lines 276, 277, 299, 309, 316, 322, 345, 356, 357, 369, 374, 385: Spodoptera in italic
Lines 283, 287, 350, 362, 363, 379: S. exigua in italic
Lines 284, 349, 362, 381: S. litura and S. littoralis in italic, i.e., S. litura and S. littoralis
Lines 285, 287, 362, 381: S. eridania and S. cosmioides in italic, i.e., S. eridania and S. cosmioides
R10. The authors acknowledge the reviewer’s comment on italized font missing on species names throughout the text. Despite being a formatting error during the submission process, a thorough revision of the manuscript has been done to include all italics properly regarding genus/species names.
- Discussion: This chapter is well-written, with only minor adjustments.
Q11. Line 537: delete fall armyworm
R11. Fall armyworm was deleted as recommended at L891.
Q12. Line 547: change fall armyworm to S. frugiperda
R12. As per reviewer recommendation, the term fall armyworm has been replaced with S. frugiperda at L911, and throughout the text.
Q13.Line 563-564: S. exigua in italic
R13. At L927, scientific names have been properly changed to italic font, and throughout the text, as per reviewer comments.

Reviewer 2 Report
Comments and Suggestions for Authors
Review
Manuscript ID 3899740 “Genetic characterization and mating disruption in Spodoptera species focusing on Spodoptera frugiperda: a systematic review"
In this review the authors summarized studies on genetic identification of Spodoptera species and the use of attractive synthetic substances for their control. The aim was to highlight advances and gaps in genetic characterization and pheromone-based mating disruption of Spodoptera species.
Please change the title. I would recommend to use as follows: Genetic characterization and mating disruption, case study on Spodoptera frugriperda: a systematic review (or a similar one).
Simple summary: The English has to be improved, for example the terminologies used in line 21-23 (understanding the pest in local condition) is not a valid termilogy in pest management and also in lines 25-27. Please clearify highlights. The authors should clarify terminologies and meanings all over the text.
Abstract: Line 31-32 does not make any sense, in some cases I have the impression that the text has parts created by AI. I will not comment in my next section the language of the paper, but I recommend a full review from this point.
2. Material methods
Section 2.1: What does it means the review search strategy? This title does not sound appropriate. Please specify at the beginning what was the overlapping between the three databases used for meta-analysis? The CABI digital library contains mostly paper abrstacts, so what was the main reason using the database, and why not only the Web of Science extended database (Master Journal List) were analysed? The search data are restricted to September 2024, why the 2025 data are not included? Lines 143-144 again not make any sense, the authors mentioned that the search was comprehensive without restriction of publication period, but the last data is from September 2024.
Section 2.2: Criteria for selecting sources: At exclusion criteria, please explain how can it be that the authors get articles not related to Spodoptera, but also mentioned keywords Spodoptera.. Pleased clarify the whole chapter and explain exactly the reasons.
Based on Figure 1 please explain how you got more papers on Scopus comparing with the Web of Science? Please increase the resolution of the Figure 1. Please also exclude the Box with removed articles.
Data analysis: Please explain how the R-softare and QGIS where used exactly on your data? It is clear QGISC was used for map illustration, this can be explained with figure 2 B, however the Figure 2A was manually created and it is very hard to follow, the grey colour used for Spodoptera are overlapping and the big issue is seeming only one species is mentioned from
Europe (It is hard to believe that only 3 papers were published in this Topic from Europe). The whole figure A2 has to be reorganised or can simply be omitted.
For Figure 3 and Figure 4: the link between Chord and Sankey diagrams are hard to follow, why not only Sankey were used in both cases. Additionally please resynchronise the colours in both diagrams, i.e., the orange for S.lutira 5 showing towards another colour and is not clear how to follow the information’s. The Chord diagram in its present form it looks like a labyrinth game. The same problem in the Figure 5.
Section 3.2.2: Please use Italics in the whole text in case of the scientific names.
Section 3.2.4: The authors mentioning the impact of genetic variability according to the geographical origins, however only 3 papers were found from Europe. How could you compare with America and 49 articles? These irrelevant comparisons representing the whole manuscript.
Sincerely,
Author Response
Alterations are highlighted with track changes in the resubmitted manuscript, along with a clean version of the revised manuscript, and point-by-point responses to all comments are provided below. The comments made the Reviewers are in bold font and our responses are in normal font.
Comments and Suggestions for Authors
Review
Manuscript ID 3899740 “Genetic characterization and mating disruption in Spodoptera species focusing on Spodoptera frugiperda: a systematic review"
In this review the authors summarized studies on genetic identification of Spodoptera species and the use of attractive synthetic substances for their control. The aim was to highlight advances and gaps in genetic characterization and pheromone-based mating disruption of Spodoptera species.
Q1. Please change the title. I would recommend to use as follows: Genetic characterization and mating disruption, case study on Spodoptera frugriperda: a systematic review (or a similar one).
R1. We thank the reviewer for the suggestion, as it better reflects the content of this study. Considering the suggestions from both reviewers 1 and 2, the title was changed to: “Genetic characterization and mating disruption, a case study on Spodoptera frugiperda (Lepidoptera: Noctuidae): a systematic review”
Q2. Simple summary: The English has to be improved, for example the terminologies used in line 21-23 (understanding the pest in local condition) is not a valid termilogy in pest management and also in lines 25-27. Please clearify highlights. The authors should clarify terminologies and meanings all over the text.
R2. We thank the reviewer for the constructive comments regarding the clarity and consistency of English terminology throughout the manuscript. We have thoroughly revised the entire text to improve language quality and ensure technical accuracy in pest management terminology.
In Simple summary section, the sentence “understanding the pest in local condition” was changed to (L24) “understanding the pest’s biology, ecology, and behavior in the different environmental conditions.”; and also at the former L25-27 and at the revised version at L28-L29, terminology has been revised from “adapt management strategies to local environments” to “adapt management strategies to local agroecological conditions”.
In accordance with this clarification, and to harmonize the terminology, in the Abstract section at L59-L60, terminology has been updated from “adapt management strategies to local conditions” to “adapt management strategies to local agroecological conditions”.
As recommended, terminology has been consistently revised throughout the manuscript to maintain scientific accuracy and clarity, with standard terminology used in integrated pest management and applied entomology.
Q3. Abstract: Line 31-32 does not make any sense, in some cases I have the impression that the text has parts created by AI. I will not comment in my next section the language of the paper, but I recommend a full review from this point.
R3. We thank the reviewer for the comment. The authors acknowledge that some sections required further language refinement and confirm that the manuscript was written without the use of AI text-generation tools. Accordingly, the entire manuscript has been carefully revised to improve clarity, coherence, and consistency of language.
Specially, regarding the text highlighted by the reviewer in the Abstract L31-L32, in the revised version, it was rephrased in terms of language to improve clarity and can be read at L34-L36 as follows: “Mating disruption through the application of synthetic pheromones has emerged as a viable alternative method for lepidopteran pest management.”
Material methods
Q4. Section 2.1: What does it means the review search strategy? This title does not sound appropriate.
R4. We thank the reviewer for this suggestion, and in the revised manuscript, in the methods section, the subsection title “2.1. Review search strategy” has been renamed to (L165) “2.1. Database search strategy”, to better reflect the content of the search strategy using the three literature databases. Also, new text has been included to justify the databases used following Editor comments, and can be read as follows at L179-L183: “The search was conducted across three online databases: Scopus (https://www.scopus.com/search/form.uri?display=advanced, 16-19 September 2024), Web of Science (https://www.webofscience.com/, 14-15 September 2024), and CABI Digital Library, the subject-specific database on agriculture relevant to the research topic under study (https://www.cabidigitallibrary.org/search/advanced/, 10-13 September 2024).”
Q5. Please specify at the beginning what was the overlapping between the three databases used for meta-analysis?
R5. We thank the reviewer for pointing out this important comment. To clarify the overlap between the three databases used for the meta-analysis, the text has been revised at the Subsection “2.2. Criteria for selecting sources and data curation workflow” in Methods section, now renamed at the revised version to “2.2. Screening and data curation workflow: inclusion and exclusion criteria of relevant studies” (L201- L202). Additionally, information on database overlap has been included at the beginning of this subsection (L203–L210) as follows: “During literature search, the initial identification resulted in a significant number of articles from each database, which were subsequently refined by removing publications that were not journal articles, such as book chapters or proceedings. After records assemblage from the three databases, duplicate entries—defined as articles retrieved simultaneously from the different databases—were identified and removed using the Microsoft Excel tool [38]. This step was essential to ensure that each database contributed with unique records not captured by the others. The overlap between databases was quantified, as illustrated in a Venn diagram (Fig. S1, provided Supplementary).”
Q6. The CABI digital library contains mostly paper abstracts, so what was the main reason using the database, and why not only the Web of Science extended database (Master Journal List) were analysed?
R6. We thank the reviewer for this important observation. Following PRISMA recommendations, we used multiple databases (CABI Abstracts database, Scopus and Web of Science) to maximize the retrieval of relevant literature and reduce potential publication bias associated with using a single database. Each database indexes different journals and document types; therefore, combining them provides a complete and more representative dataset for the review. The CAB Abstracts (CABI) database was selected for its strong coverage of applied biological and agricultural sciences, ensuring the inclusion of relevant studies not indexed in general scientific databases. Thus, to address the reviewer´s comment, in the subsection “2.1. Database search strategy”, new text has been included to justify the databases used, and can be read at L179-L183: “The search was conducted across three online databases: Scopus (https://www.scopus.com/search/form.uri?display=advanced, 16-19 September 2024), Web of Science (https://www.webofscience.com/, 14-15 September 2024), and CABI Digital Library, the subject-specific database on agriculture relevant to the research topic under study (https://www.cabidigitallibrary.org/search/advanced/, 10-13 September 2024).”
Although CABI mainly provides abstracts, we used these records during the title and abstract initial screening stage. As detailed in the supplementary material (Table S1) and the PRISMA flow diagram (Figure 1) included in the Methods section, CABI contributed with unique and relevant records not retrieved from Scopus or Web of Science, confirming its added value to our search strategy.
Despite CABI database provide mostly abstracts, next text regarding articles’ accessibility has been included in the Methods section subsection 2.2. Screening and data curation workflow: inclusion and exclusion criteria of relevant studies” L231-236: “For retrieved records, the full text was obtained directly from the databases (i.e., open access articles) or by alternative sources (e.g., publisher websites, b-on - https://www.b-on.pt, institutional repositories, or directly from the corresponding author). For records indexed at CABI Abstracts, the full texts were obtained through the CABI Library institutional voucher account access.”
Q7. The search data are restricted to September 2024, why the 2025 data are not included? Lines 143-144 again not make any sense, the authors mentioned that the search was comprehensive without restriction of publication period, but the last data is from September 2024.
R7. We thank the reviewer for this observation. The literature search was completed in September 2024, which served as the cut-off date for data inclusion in this review. Considering the extensive methodology applied according to PRISMA guidelines, which involved a thorough manual curation of several articles individually and manually to assess their eligibility for screening, data analyses and interpretation, along with manuscript preparation and writing process, September 2024 was set out for article inclusion, which includes all available articles published until this date. However, we acknowledge the reviewer’s comment that additional relevant studies may have been published since then and should be considered in future reviews. We believe that the systematic review performed gives an important yet lacking contribution to the body of knowledge on Spodoptera species, and in particular to S. frugiperda, regardingthe genetic identification and the use of attractive synthetic substances for their control, showing that genetic information helps to better understand these pests, but the effectiveness of such control methods has been variable.
Q8. Section 2.2: Criteria for selecting sources: At exclusion criteria, please explain how can it be that the authors get articles not related to Spodoptera, but also mentioned keywords Spodoptera. Pleased clarify the whole chapter and explain exactly the reasons.
R8. To improve greater detail on the transparent inclusion and exclusion criteria applied at the systematic review, the Subsection “2.2. Criteria for selecting sources and data curation workflow” in the Methods section have been renamed to “2.2. Screening and data curation workflow: inclusion and exclusion criteria of relevant studies” and rewritten to accommodate the editor and reviewers’ comments, and can be read as follows at L201- L238:
“2.2. Screening and data curation workflow: inclusion and exclusion criteria of relevant studies
During literature search, the initial identification resulted in a significant number of articles from each database, which were subsequently refined by removing publications that were not journal articles, such as book chapters or proceedings. After records assemblage from the three databases, duplicate entries—defined as articles retrieved simultaneously from the different databases—were identified and removed using the Microsoft Excel tool [38]. This step was essential to ensure that each database contributed with unique records not captured by the others. The overlap between databases was quantified, as illustrated in a Venn diagram (Fig. S1, provided Supplementary).
After, the screening process by manual curation was carried out through title relevance and assessment of both abstracts and full-texts content, applying inclusion and exclusion criteria. The inclusion criterion required only original research articles related to genetic and pheromone-based control on Spodoptera species. Two exclusion criteria were subsequently applied. The first exclusion criterion involved the removal of articles that did not meet the topics under study, namely: (i) articles not related to Spodoptera species (e.g. other insect species or taxa); (ii) articles not relevant to the main topics of this review (e.g., biological control, insecticide resistance, or chemical management); and (iii) removal of reviews articles. The second exclusion criterion was based on the assessment of the abstract and full text content of each article by evaluating the relevance of the information to the topics under study. Thus, (i) articles with inaccessible full text or abstracts lacking sufficient data for analysis; (ii) articles with abstracts or full texts in languages whose data were not accessible; and (iii) other article types not identified during the first exclusion (e.g., reports, proceedings), were eliminated for subsequent data processing. For retrieved records, the full text was obtained directly from the databases (i.e., open access articles) or by alternative sources (e.g., publisher websites, b-on - https://www.b-on.pt, institutional repositories, or directly from the corresponding author). For records indexed at CABI Abstracts, the full texts were obtained through the CABI Library institutional voucher account access. After applying the two exclusion criteria, the remaining articles were considered eligible for inclusion in the review (Fig. 1). The details of the eligible articles included in the systematic review are provided in the Supplementary Table S2.”
Particularly, regarding reviewer’s comment on retrieving papers not related to Spodoptera despite the inclusion of Spodoptera as a keyword, these results are mainly attributed to the databases also retrieved articles related to other insect species or taxa, by applying the Boolean operators (Table 1) for each database. In some cases, keywords appeared in broader contexts, such as titles, abstracts, or reference lists, resulting in irrelevant records. These articles were consequently excluded, as they did not cover Spodoptera species. On the other hand, articles that focused on Spodoptera but addressed unrelated topics (e.g., biological control, insecticide resistance, or chemical management) were also excluded to ensure that the final dataset included only studies directly relevant to the objectives of this review.
Q9. Based on Figure 1 please explain how you got more papers on Scopus comparing with the Web of Science?
R9. We thank the reviewer for this observation. The higher number of records retrieved from Scopus compared to Web of Science is mainly explainable by differences in database coverage and indexing practices. Both Scopus and Web of Science index a large number of publications, but our results may reflect the inconsistencies in indexing practices in different databases, highlighting the need to include different databases when performing a comprehensive review.
At the initial phase, our search included filters within each database, covering document type (e.g., book chapters, reviews) and language. Although these filters were applied to select records according to our inclusion criteria, some publications were not automatically filtered due to differences in how records are indexed in each of the three databases. These clarifications have been included in the subsection 2.1, (L185-1940): “However, this review only included articles in English language, both full-text and abstract, articles with the full-text available, and publication type focusing on original research articles relevant to the review topics. Search queries were organized into three categories: 1) Genetic characterization and strain identification in Spodoptera species; 2) Sex pheromones and mating disruption in Spodoptera species and 3) Effectiveness of sex pheromones for mating disruption in relation to genetic characterization and strain identification in Spodoptera species. For each category, basic and specific keywords were used, as listed in Table 1. Boolean operators (AND/OR) and truncation symbols were combined with these terms to formulate the search queries.”
Q10. Please increase the resolution of the Figure 1.
R10.The resolution of Figure 1 has been increased, and the updated figure has been included (L240) in the revised manuscript, as recommended.
Q11. Please also exclude the Box with removed articles.
R11. We thank the reviewer for this observation, however, considering that the methodology followed the PRISMA guidelines methodology and the PRISMA 2020 flow diagram for new systematic reviews, which included searches of databases and registers only available in https://www.prisma-statement.org/prisma-2020-flow-diagram, according to this template, the number of articles excluded and included should be indicated at each stage of the process. The PRISMA flow diagram shows the information through the different phases of a systematic review. It maps out the number of records identified, included and excluded, along with the details for exclusions.
Q12. Data analysis: Please explain how the R-softare and QGIS where used exactly on your data? It is clear QGISC was used for map illustration, this can be explained with figure 2 B, however the Figure 2A was manually created and it is very hard to follow, the grey colour used for Spodoptera are overlapping and the big issue is seeming only one species is mentioned from Europe (It is hard to believe that only 3 papers were published in this Topic from Europe).
The whole figure A2 has to be reorganised or can simply be omitted.
R12. R software was originally used to produce the chord diagrams. However, in the current version of the manuscript, these diagrams have been replaced by Sankey diagrams, and R is no longer used in the final analysis. QGIS, as correctly noted, was employed to produce the map presented in Figure 2. Thus, the citation of R in the text and the corresponding reference in the final list were removed. In this regard, the reference numbering has been updated throughout the text and in the final list.
Also, following Reviewer’s suggestion, Figure 2A has been removed, and only Figure 2B remains in the revised version as Figure 2.
Indeed, the limited number of European studies is consistent with the current distribution of the most studied species, Spodoptera frugiperda, as highlighted in this review. This species is native to the Americas and has only recently been reported in Europe, where it is not yet widely established. Thus, articles retrieved after the PRISMA methodology from the 3 databases at that time of screening (September 2024) were used for the review, despite only 3 were accounted for Europe. To counteract the reviewer comment, a new sentence has been included in the Results Section, subsection 3.2.4., L7472-750 and reads as follows: “Most existing studies originate from the Americas, the pest’s native range, while relatively few have been conducted in regions where the pest has been recently reported, such as Europe. This imbalance in research efforts poses a major limitation for making direct comparisons of its impacts across regions overall.”
Q13. For Figure 3 and Figure 4: the link between Chord and Sankey diagrams are hard to follow, why not only Sankey were used in both cases. Additionally please resynchronise the colours in both diagrams, i.e., the orange for S. lutira 5 showing towards another colour and is not clear how to follow the information’s. The Chord diagram in its present form it looks like a labyrinth game. The same problem in the Figure 5.
R13. As suggested, the chord diagrams have been replaced by Sankey diagrams to improve clarity and readability, in Figures 3 and 5. The color scheme has been carefully revised to ensure consistency across all figures (Figures 3, 4 and 5), maintaining the same color for each species.
Q14. Section 3.2.2: Please use Italics in the whole text in case of the scientific names.
R14. All scientific names have been corrected in italics throughout the text, in accordance with the observations also pointed out by reviewer 1.
Q15. Section 3.2.4: The authors mentioning the impact of genetic variability according to the geographical origins, however only 3 papers were found from Europe. How could you compare with America and 49 articles? These irrelevant comparisons representing the whole manuscript.
R15. We thank the reviewer for this observation. Indeed, the limited number of European studies is consistent with the current distribution of the most studied species, Spodoptera frugiperda, as highlighted in this review. This species is native to the Americas and has only recently been reported in Europe, where it is not yet widely established. Our review aims not only to compare genetic studies in Spodoptera species between regions but also to highlight advances in studies on this topic, including methodological developments, knowledge gaps, and emerging strategies for mating disruption.
In the subsection 3.2.4, the objective of the discussion is the impact of genetic variability on the effectiveness of pheromones for mating disruption, and the effectiveness according to pheromonal composition or origin. To date, genetic variability is only known in the Spodoptera frugiperda, with most studies still referring to its region of origin, the Americas, as evidenced in this review. However, to clarify these points in emphasizing that conclusions about low European studies while most of the evidence comes from studies in the Americas, pest’s native region, new text have been added at 747L-L750 and reads as follows: “Most existing studies originate from the Americas, the pest’s native range, while relatively few have been conducted in regions where the pest has been recently reported, such as Europe. This imbalance in research efforts poses a major limitation for making direct comparisons of its impacts across regions overall.”

Reviewer 3 Report
Comments and Suggestions for Authors
The manuscript under review fully complies with the scope and focus of the Insects.
The Abstract and Summary also meet the requirements and reflect the relevance, methodology, and results of the literature review.
The review is well illustrated, and there are no comments regarding the formatting of the tables and illustrations.
Overall, the review addresses a relevant topic and is of interest not only to general entomology but also to the development of methods for controlling the reproduction of common insect species. Representatives of the genus Spodoptera are serious agricultural pests in many regions of the world. Therefore, the effective development of control methods requires precise knowledge of the taxonomic position of a given species and an assessment of its life strategy and developmental biology.
A particular strength of this work is the use of the PRISMA method and protocol, combined with a modern office and statistical software package (Microsoft Excel, R software, QGIS). This allowed the authors to obtain statistically reliable data and visualize them effectively.
I would like to point out the following comments:
- "PRISMA" should be added to the keywords.
- I would strongly recommend that authors provide the R script used to summarize the data in an appendix to the manuscript, as this would allow future readers to summarize the data by other parameters and thus increase the value of this Review. Currently, open access to R scripts and codes allows for standardization of research and makes it more open and verifiable, which is why many highly ranked journals publish them in appendices to the manuscript.
Author Response
Alterations are highlighted with track changes in the resubmitted manuscript, along with a clean version of the revised manuscript, and point-by-point responses to all comments are provided below. The comments made the Reviewers are in bold font and our responses are in normal font.
Comments and Suggestions for Authors
The manuscript under review fully complies with the scope and focus of the Insects.
The Abstract and Summary also meet the requirements and reflect the relevance, methodology, and results of the literature review.
The review is well illustrated, and there are no comments regarding the formatting of the tables and illustrations.
Overall, the review addresses a relevant topic and is of interest not only to general entomology but also to the development of methods for controlling the reproduction of common insect species. Representatives of the genus Spodoptera are serious agricultural pests in many regions of the world. Therefore, the effective development of control methods requires precise knowledge of the taxonomic position of a given species and an assessment of its life strategy and developmental biology.
A particular strength of this work is the use of the PRISMA method and protocol, combined with a modern office and statistical software package (Microsoft Excel, R software, QGIS). This allowed the authors to obtain statistically reliable data and visualize them effectively.
I would like to point out the following comments:
The authors would like to thank the reviewer for the positive comments.
Q1. 1. "PRISMA" should be added to the keywords.
R1. “PRISMA” has been included in the keywords as recommended.
Q2. 2. I would strongly recommend that authors provide the R script used to summarize the data in an appendix to the manuscript, as this would allow future readers to summarize the data by other parameters and thus increase the value of this Review. Currently, open access to R scripts and codes allows for standardization of research and makes it more open and verifiable, which is why many highly ranked journals publish them in appendices to the manuscript.
R2. Thank you for the important comment. Considering that R software/script was originally used for generating the chord diagrams, and that these diagrams have been replaced by Sankey diagrams as per reviewer 2 suggestions (Figures 3 and 5) in the revised version of the manuscript, R is no longer used in the final analysis. Thus, the R script was not included, and the R software was removed from the text, with all references numbering updated throughout the manuscript accordingly.

Round 2
Reviewer 2 Report
Comments and Suggestions for Authors
Review-Revised version
Manuscript ID 3899740 “Genetic characterization and mating disruption in Spodoptera species, a case study on Spodoptera frugiperda (Lepidoptera, Noctuidae): a systematic review "
This manuscript presents a systematic review of genetic characterization and pheromone-based mating disruption in Spodoptera species, focusing on S. frugiperda. The topic is highly relevant for pest management and agricultural biosecurity, especially given the global spread of S. frugiperda. The study follows PRISMA guidelines, the revised version shows clear improvement over the original submission: methods are more transparent, figures better presented, and terminology standardized. However, some areas still require strengthening before publication.
The majority of studies addressed genetic characterization, while few integrated pheromone and genetic approaches. The pheromone control results are inconsistent across regions.
Introduction: Well-organized and improved, but somewhat redundant in pest biology details.
Conclusion: The authors should include a future outlook linking genomics and chemical researches regarding to IPM programs.
Further English polishing recommended, please verify reference formatting and ensure consistency.
The revised manuscript demonstrates significant improvements and addresses my main concerns. However, to reach the publication standard expected for a systematic review, the authors should enhance synthesis in the Discussion, improve reproducibility transparency (search strings, validation of data extraction), and refine the language for conciseness. If these are addressed satisfactorily, the paper would merit acceptance after Major revision.
Sincerely,

Author Response
Reviewer 2
Review-Revised version
Manuscript ID 3899740 “Genetic characterization and mating disruption in Spodoptera species, a case study on Spodoptera frugiperda (Lepidoptera, Noctuidae): a systematic review "
This manuscript presents a systematic review of genetic characterization and pheromone-based mating disruption in Spodoptera species, focusing on S. frugiperda. The topic is highly relevant for pest management and agricultural biosecurity, especially given the global spread of S. frugiperda. The study follows PRISMA guidelines, the revised version shows clear improvement over the original submission: methods are more transparent, figures better presented, and terminology standardized. However, some areas still require strengthening before publication.
The majority of studies addressed genetic characterization, while few integrated pheromone and genetic approaches. The pheromone control results are inconsistent across regions.
Introduction:
Q. Well-organized and improved, but somewhat redundant in pest biology details.
R. The authors appreciate the reviewer’s observation. To address this point, we have carefully revised the Introduction, particularly, to remove redundant details on pest biology and streamline the content for greater focus and conciseness. The revised text regarding to pest biology and its global impacts at lines 75–89:
“Spodoptera frugiperda is recognised as major pest threatening to agriculture worldwide, causing significant damage to important crops such maize and rice, particularly in newly invaded regions. It poses a critical challenge to food security, especially in areas where maize is a staple crop and resources for pest management are limited, as is often the case in the African context. Annual losses caused by this pest on maize production crop are estimated at US$300–500 million in the U.S., US$1.1–4.7 billion in sub-Saharan Africa (up to US$13 billion across major crops), and 4.1–17.7 million tons only in 12 key African countries [6-9]. Its destructive capacity is due to its biological characteristics, which include a wide host range, high migratory capacity, strong fertility, and the absence of a diapause mechanism, i.e., population outbreaks largely depend on environmental conditions, in addition to their ability to develop resistance to pesticides [9-12]. While chemical control remains, the main method used against S. frugiperda, itseffectiveness is limited, with reduced susceptibility and field-evolved resistance reported in both native and invaded regions [13-15]. Resistance to Bacillus thuringiensis (Bt) toxins has also emerged despite the widespread use of transgenic Bt crops for S. frugiperda control [15].“
Conclusion:
Q. The authors should include a future outlook linking genomics and chemical researches regarding to IPM programs.
R. We thank the reviewer for this valuable suggestion. We have revised the text of the Conclusions section, highlighting this point by incorporating an outlook and the importance of integrating genomic and chemical research for advancing IPM programs. The included text at the revised manuscript reads as follows at Lines 921-944:
“Understanding the dispersal behaviour, reproductive biology, and population structure of S. frugiperda is essential to ensure effective population reduction. Incorporating molecular tools to identify genetic variation and strain composition across different agroecological contexts could support more precise pest management efforts. Pheromone-based mating disruption adapted to the specific agroecological context, could enhance integrated pest management programs by disrupting the pest’ reproduction, reducing population pressure over time, and enabling compatibility with other control methods such as biological control and chemical control (preferably with selective insecticides), thereby minimizing environmental impact and pest resistance to pesticides.
The pest migratory behaviour and the potential introduction of new invasive lineages or hybridization highlight the importance of continuous genetic monitoring and locally adapted management strategies. Integrating population genetic studies with semiochemicals, particularly the most effective pheromone blends, offers a promising approach for sustainable pest control. Further research should focus on: (1) mapping pheromone components and its proportion with their genetic determinants across populations, validating their effects on behavioural responses; (2) characterizing pheromone production and response in local populations; and (3) developing pheromone-based mating disruption tailored to local populations, potentially combined with host-plant resistance and biological control to enhance integrated pest management.
In conclusion, this research underscores the relevance for a comprehensive and interdisciplinary approach to the management of S. frugiperda, particularly in sub-Saharan Africa, where its rapid spread is significantly impacting food security. Accurate genetic characterization is essential for strain identification and understanding population dynamics, which in turn supports effective control strategies.”
Q. Further English polishing recommended, please verify reference formatting and ensure consistency.
R. We appreciate the recommendations. Thus, to ensure and improve the quality of technical English and consistency, the manuscript has undergone a thorough English polishing made by the authors and further performed by two senior external experts in the area, that ensured language grammar conciseness and clarity, which have been properly recognized at the acknowledgments section.
Also, all references have been fully formatted according to the reference style indicated in the journal’s guidelines and template, ensuring consistency throughout the reference list.
Q. The authors should enhance synthesis in the Discussion, improve reproducibility transparency (search strings, validation of data extraction), and refine the language for conciseness.
R. We sincerely thank the reviewer for the insightful and constructive feedback. In response, we have further strengthened the Discussion section to provide a more integrated and cohesive synthesis of the findings. Revised manuscript at the Discussion has been overall improved by highlighting general insights that emerged from the dataset obtained from the systematic review, connecting results to the objectives of this review, and integrating insights from genetic and pheromone-based research.
Also, we appreciate the reviewer’s comment highlighting the importance of transparency and reproducibility. As indicated the data extraction was conducted using a standardized Excel sheet, provided as supplementary material. However, to enhance transparency and reproducibility, a clarification was added at the beginning of the Materials and Methods section, 2.3. Data extraction and analysis section, indicating that data extraction was performed systematically using a standardized Excel sheet, verified independently by two reviewers, and cross-checked for reproducibility. All extracted variables and their corresponding sources are available in Supplementary Table S3. This revision ensures that readers and future researchers can verify and reproduce the results, in line with PRISMA 2020 recommendations. The revised text now reads at Materials and Methods, Lines 372-376:
“Data extraction was performed using a standardised Excel sheet to ensure consistency across studies. A subset of records was checked by two reviewers independently to verify the accuracy and standardisation of the extracted data and validate reproducibility. The resulting table included general information such as the article title, authors, year of publication, country, region of study, Spodoptera species studied, and associated host crops.”
Rewarding search strings, all keywords used for conducting the searches for relevant articles at the three databases are available at Table 1 (L168), as well as the Boolean operators, together with search codes and results retrieved from each database are available at the Supplementary Table S1. Also, to ensure reproducibility transparency, all this important information is mentioned at the Material and Methods Section at Lines 317–323:
“For each category, basic and specific keywords were used, as listed in Table 1. Boolean operators (AND/OR) and truncation symbols were combined with these terms to formulate the search queries. The search codes and results for each database are provided in Table S1, in supplementary material, and a table with all articles identified was published at the online repository Figshare [36] and all references managed in Mendeley software [37].”
Additionally, the manuscript has undergone a thorough language refinement to improve clarity, conciseness, and readability, ensuring it meets the high standards expected for a systematic review.
Round 3
Reviewer 2 Report
Comments and Suggestions for Authors
Second Review-Revised version
Manuscript ID 3899740 “Genetic characterization and mating disruption in Spodoptera species, a case study on Spodoptera frugiperda (Lepidoptera,Noctuidae): a systematic review"
The revised shows improvements in structure, methodological clarity, and English language quality compared to the earlier version. The authors have addressed most of the reviewers’ main concerns effectively:
- Introduction: Redundant biological details were removed, and the background is now more concise and focused on the review’s aims.
- Methods: Transparency and reproducibility have been notably enhanced.
- Discussion and Conclusion: The revised version offers improved synthesis, integrating findings on genetic characterization and pheromone-based management. The conclusion now explicitly connects genomic research with semiochemical applications for Integrated Pest Management (IPM), fulfilling my request.
- Minor comments:
- Ensure consistent terminology (“sex pheromones” or “sexual pheromones”)-choose one.
- Review sentence flow in the abstract and discussion for conciseness.
- English usage has been significantly improved, though occasional awkward phrasing remains (examples: line 60, such maize and rise – such as..,, line 91 control certain Lepidoptera species – control of certain…. etc).
Sincerely,
Author Response
Reviewer 2 Comments
The revised shows improvements in structure, methodological clarity, and English language quality compared to the earlier version. The authors have addressed most of the reviewers’ main concerns effectively:
Introduction: Redundant biological details were removed, and the background is now more concise and focused on the review’s aims.
Methods: Transparency and reproducibility have been notably enhanced.
Discussion and Conclusion: The revised version offers improved synthesis, integrating findings on genetic characterization and pheromone-based management. The conclusion now explicitly connects genomic research with semiochemical applications for Integrated Pest Management (IPM), fulfilling my request.
R. The authors appreciate this reviewer’s positive assessment of the revised manuscript and confirm that all previous comments have been carefully addressed and integrated in a constructive manner to further improve the quality of the paper. The authors have also included acknowledgements to the reviewers for their valuable and constructive suggestions in the Acknowledgements section.
Minor comments:
Q. Ensure consistent terminology (“sex pheromones” or “sexual pheromones”)-choose one.
R. The authors thank the reviewer for this observation. A general review was conducted, and all instances of the term “sex” were harmonized to “sexual” at Line 35 (Abstract) and also at Table 1 of the Methods section, the missing truncation was added for the keyword “sex*,” and the corresponding caption was included in the table title as “*sexual.”
Q. Review sentence flow in the abstract and discussion for conciseness.
R. The authors thank the reviewer for this constructive suggestion. The abstract and discussion sections have been carefully revised to improve sentence flow and ensure conciseness. Long or redundant sentences were shortened, transitions refined, and key ideas made more direct to enhance readability while preserving the scientific accuracy of the text.
Abstract: Minor revised at the sentences provided at Line 34-36, Line 42-46 and Line 49
Discussion: Minor revised at the sentences provided at Line 464-465, Line 590-591 and Line 599
Q. English usage has been significantly improved, though occasional awkward phrasing remains (examples: line 60, such maize and rise – such as..,, line 91 control certain Lepidoptera species – control of certain…. etc).
R. The authors thank the reviewer for the careful attention to language refinement. Specifically, text at L64 and L95 was corrected as suggested. Additionally, some minor phrasings were adjusted to improve English fluency throughout the manuscript: L466 and L593.
These changes enhance the readability and overall linguistic quality of the manuscript.